# WHIRLY1 Acts Upstream of ABA-Related Reprogramming of Drought-Induced Gene Expression in Barley and Affects Stress-Related Histone Modifications

**DOI:** 10.3390/ijms24076326

**Published:** 2023-03-28

**Authors:** Minh Bui Manh, Charlotte Ost, Edgar Peiter, Bettina Hause, Karin Krupinska, Klaus Humbeck

**Affiliations:** 1Institute of Biology, Martin Luther University Halle-Wittenberg, Weinbergweg 10, 06120 Halle, Germany; 2Plant Nutrition Laboratory, Institute of Agricultural and Nutritional Sciences, Faculty of Natural Sciences III, Martin Luther University Halle-Wittenberg, 06120 Halle, Germany; 3Department of Cell and Metabolic Biology, Leibniz Institute of Plant Biochemistry, Weinberg 3, 06120 Halle, Germany; 4Institute of Botany, Christian-Albrechts-University (CAU), 24098 Kiel, Germany

**Keywords:** abscisic acid, drought stress, histone modifications, *Hordeum vulgare*, transcriptome, WHIRLY

## Abstract

WHIRLY1, a small plant-specific ssDNA-binding protein, dually located in chloroplasts and the nucleus, is discussed to act as a retrograde signal transmitting a stress signal from the chloroplast to the nucleus and triggering there a stress-related gene expression. In this work, we investigated the function of WHIRLY1 in the drought stress response of barley, employing two overexpression lines (oeW1-2 and oeW1-15). The overexpression of *WHIRLY1* delayed the drought-stress-related onset of senescence in primary leaves. Two abscisic acid (ABA)-dependent marker genes of drought stress, *HvNCED1* and *HvS40*, whose expression in the wild type was induced during drought treatment, were not induced in overexpression lines. In addition, a drought-related increase in ABA concentration in the leaves was suppressed in *WHIRLY1* overexpression lines. To analyze the impact of the gain-of-function of WHIRLY1 on the drought-related reprogramming of nuclear gene expression, RNAseq was performed comparing the wild type and an overexpression line. Cluster analyses revealed a set of genes highly up-regulated in response to drought in the wild type but not in the WHIRLY1 overexpression lines. Among these genes were many stress- and abscisic acid (ABA)-related ones. Another cluster comprised genes up-regulated in the oeW1 lines compared to the wild type. These were related to primary metabolism, chloroplast function and growth. Our results indicate that WHIRLY1 acts as a hub, balancing trade-off between stress-related and developmental pathways. To test whether the gain-of-function of WHIRLY1 affects the epigenetic control of stress-related gene expression, we analyzed drought-related histone modifications in different regions of the promoter and at the transcriptional start sites of *HvNCED1* and *HvS40*. Interestingly, the level of euchromatic marks (H3K4me3 and H3K9ac) was clearly decreased in both genes in a *WHIRLY1* overexpression line. Our results indicate that WHIRLY1, which is discussed to act as a retrograde signal, affects the ABA-related reprogramming of nuclear gene expression during drought via differential histone modifications.

## 1. Introduction

Due to global climate anomalies, drought stress is an increasing, worldwide threat to plant growth and causes yield depressions in our major crops and thereby socioeconomic crises [1,2]. Drought results in decreased water availability in cells which finally leads to a collapse of cellular functions that require water. The early signs are a cessation of cell elongation and division reflected by a stop in growth, decreased biochemical processes, such as photosynthesis, a loss of turgor evident via wilting and later on also the stress-related onset of leaf senescence.

However, plants have evolved sophisticated mechanisms to cope with such unfavorable conditions, e.g., changes in root architecture or lowering the water potential of the cells via biosynthesis of compatible solutes, such as carbohydrates or proline, promoting the uptake of water. Understanding these complex processes of acclimation to water deprivation is of high interest to scientists, breeders and agriculturists to ensure yield under uncertain conditions of water shortage. One major upstream regulator mediating drought stress responses is phytohormone abscisic acid (ABA), regulating the stress-related reprogramming of gene expression besides stomatal closure [3,4]. ABA originates from the epoxy-xanthophylls violaxanthin and neoxanthin. The key rate-limiting step in its biosynthesis is the cleavage to the C-15 fragment xanthoxin via 9-*cis*-epoxycarotenoid dioxygenase (NCED) [5,6,7]. The first steps of ABA biosynthesis are located in the chloroplast, which by itself is very sensitive to water deprivation and acts as a major sensor of stress [8,9,10].

An increasing number of publications has revealed that stress responses in plants are also controlled via higher-order epigenetic regulatory mechanisms, including small RNAs, DNA-methylation and differential histone modifications [11,12]. Especially, ABA-signaling pathways have been shown to involve histone-modifying enzymes, e.g., histone deacetylases (HDAs) [4,13,14,15]. Recently, a plant-specific single-stranded DNA (ss-DNA)-binding protein family called WHIRLY and dually located in plastids or mitochondria and the nucleus has been described [16,17]. This multifunctional protein family, which has denominated WHIRLY due to its protein structure, is involved in plant development and plants’ responses to biotic and abiotic stress [14,18,19,20,21,22,23]. WHIRLY proteins are discussed to act as a retrograde signal and the transfer of WHIRLY1 from the chloroplast to the nucleus has been shown in transplastomic tobacco plants synthesizing an HA-tagged form of AtWHIRLY1 in the chloroplast, which was then immunologically identified in the nucleus where it changed the expression of pathogen-related genes. The findings indicated a transport of WHIRLY1 from the chloroplast to the nucleus, where it obviously functioned in transcriptional regulation [24,25]. In chloroplasts, WHIRLY proteins have been identified in the fraction of the transcriptionally active chromosome (TAC)-complex which shares many proteins with the nucleoids [26,27,28]. In the nucleus, WHIRLY proteins affect the expression of stress- and development-related genes, and an interaction of WHIRLY with the promoter of the transcription factor gene *WRKY53* was shown [18,29]. However, the exact mode of transcriptional regulation is not yet known. Interestingly, a knock-down of *WHIRLY1* can affect histone modifications in a senescence-specific target gene in barley (*Hordeum vulgare* L.) [14].

In this paper, we further investigated the function of WHIRLY1, one of the two WHIRLY proteins in barley, showing the effect of the WHIRLY1 gain-of-function on the reprogramming of gene expression during drought stress. We present a model with WHIRLY1 as a scaffold interacting with epigenetic regulators to control stress-related gene expression.

## 2. Results

### 2.1. Overexpression of HvWHIRLY1 Delays Drought-Induced Onset of Senescence

Barley plants were grown on soil at about 65% soil water capacity (control conditions) until 11 days after sowing (das). On that day, irrigation was stopped for half of the plants, while the rest (controls) were further irrigated to keep the soil water content at 65%. After withholding irrigation, the water capacity in the soil of the drought-treated plants decreased continuously: at 19 das it was reduced to about 35%, and reached almost zero at 27 das (Figure 1A). In response to this desiccation process, primary barley leaves entered premature senescence at 25 das, as indicated by a decrease in chlorophyll content (Figure 1B), and with a four day delay there was also a drastic decrease in photosystem II (PS II) efficiency (Figure 1C), whereby both parameters were markers of senescence. Developmental leaf senescence started in control plants about 4 days later.

To investigate the effect of the gain-of-function of WHIRLY1, two lines overexpressing *HvWHIRLY1* were established with about 50 (oeW1-2) or 10 (oeW1-15) times higher levels of WHIRLY1 protein [30] and even higher levels of mRNAs compared to wild-type plants (see Appendix A). In contrast to the wild type (WT), the overexpression lines showed a delay in the onset of senescence, by four (oeW1-15) or six (oeW1-2) days under stress conditions (Figure 1B,C). The chlorophyll content and PS II efficiency of wild-type and overexpression plants at 29 and 33 das are presented in Figure 1D, revealing that the overexpression of *WHIRLY1* delayed the onset of drought-stress-induced senescence in barley leaves. At 33 das, both overexpression lines showed a significantly higher PS II efficiency and significantly higher chlorophyll content than the WT, whereby oeW1-2 plants with a higher expression level of *WHIRLY1* showed more dramatic effects than oeW1-15 plants. This is obvious in Figure 1E, where representative primary leaves of control and drought-stressed WT and overexpression lines at 29 das and 33 das are displayed.

### 2.2. Overexpression of WHIRLY1 Suppressed Drought-Induced Expression of ABA-Related Genes and ABA Accumulation

Since ABA is an upstream signal of drought-induced premature senescence, the expression of some ABA-related marker genes was investigated in overexpression lines in comparison to WT plants (Figure 2). *HvNCED1*, *CCD5* and *CCD8* are known barley genes involved in the biosynthesis of ABA [6]. Seventeen days after stopping irrigation, these ABA biosynthesis genes and the ABA-dependent senescence- and stress-related gene *HvS40* were clearly induced in the WT plants. Interestingly, the overexpression of *HvWHIRLY1* suppressed this drought-related induction of ABA biosynthesis genes and *HvS40*. The expression of these genes was already significantly lower under control conditions, especially in oeW1-2 with a higher expression level of *WHIRLY1*. To test whether the altered expression of ABA biosynthesis genes impinged on ABA accumulation, we measured ABA levels during drought stress in WT and the *WHIRLY1* overexpression plants (Figure 2B). The strong accumulation of ABA in WT during drought treatment is clearly suppressed when *WHIRLY1* is overexpressed in both lines.

### 2.3. Overexpression of WHIRLY1 Affects Reprogramming of Gene Expression during Drought Stress

To investigate the effects of *WHIRLY1* overexpression on the drought-related reprogramming of nuclear gene expression, RNAseq analyses were performed comparing the transcriptome of the oeW1-2 line and WT after 17 days of drought. Differentially expressed genes (DEGs) are listed in Appendix A. In total, 4403 DEGs were identified, 2333 being down-regulated and 2070 being up-regulated specifically in the oeW1-2 line (Figure 3A), indicating a major effect of *WHIRLY1* overexpression on gene expression during drought stress. Although there were already effects of *WHIRLY1* overexpression under control conditions (Figure 3A), the number of DEGs (1464) was much lower than under drought conditions (4403). Volcano plots (Figure 3B) show that the overexpression of *WHIRLY1* under control and drought conditions causes a similar number of genes being up- and down-regulated, suggesting that WHIRLY1 acts in both directions, as a positive and a negative regulator. DEGs were clustered according to their expression pattern (Figure 3C), illustrating clear effects of *WHIRLY1* overexpression on gene expression in control and drought conditions. The upper main cluster shows that many genes which were clearly up-regulated in WT during drought conditions were not up-regulated when *WHIRLY1* was overexpressed. To classify these genes according to their function, a GO-enrichment analysis [31] was performed (Appendix A). Interestingly, these genes, not induced in response to drought stress due to *WHIRLY1* overexpression, primarily have functions in transport activities, senescence processes, hormone-related functions and biotic and abiotic stress-sensing functions, including ABA-related processes (see color codes in Appendix A). In addition, we manually selected all barley ABA-related genes and plotted their expression levels in WT and oeW1-2 in control and drought conditions as a heatmap (Figure 3D). Genes related to ABA biosynthesis, such as *NCED1*, *CCD8* and *CCD5*, encoding carotenoid cleavage dioxygenases [32], and the ß-glucosidase BG7, possibly involved in glucosidase-mediated stress-activated ABA production [33], were up-regulated in WT during drought stress, but not in line oeW1-2. These results correspond to the qRT-PCR analyses (Figure 2). Other ABA-related genes show a different behavior. *ZEP5*, encoding an ABA zeaxanthin epoxidase functioning in the first step of ABA biosynthesis [34] which is already up-regulated under the control condition in WT, is down-regulated in oeW1-2 under both control and drought conditions. On the other hand, several barley genes possibly involved in ABA metabolism (e.g., aldehyde oxidases AO3 and AO4, which possibly catalyze the final step in ABA biosynthesis [35], ß-glucosidases BG4 and BG9 and another carotenoid cleavage dioxygenase (CCD4)) were up-regulated in oeW1-2 when compared to WT. These results show that the overexpression of *WHIRLY1* affects the expression of genes related to ABA metabolism.

The lower main cluster in Figure 3C shows genes that are highly up-regulated in the overexpression line (mostly under both conditions: control and drought) when compared to WT. GO-enrichment analysis of these genes was performed (Appendix A). Interestingly, many of these genes are involved in chloroplast development and chloroplast function (marked in Appendix A). Collectively, our data indicate that WHIRLY1 in barley is a major bi-directional regulator, on the one hand suppressing the expression of genes involved in transport processes, senescence and biotic and abiotic stress responses, and on the other hand stimulating gene-sets involved in chloroplast development.

### 2.4. Overexpression of WHIRLY1 Causes Loss of Euchromatic Marks in the ABA-Related Genes HvNCED1 and HvS40

Recent work has shown that stress signaling involves epigenetic regulation, including differential histone modifications in target genes [13,36,37]. To test whether the overexpression of *WHIRLY1* affects this epigenetic control level, differential histone modifications associated with the promoters and transcriptional start sites of the two genes *HvNCED1* and *HvS40* were investigated via ChIP-qPCR. As shown before, the drought-induced expression of these two genes was suppressed when *WHIRLY1* was overexpressed. Loading with two euchromatic marks (H3K4me3 and H3K9Ac) was tested in the control and drought-treated samples of both genotypes. It is known that the establishment of euchromatic marks, especially near promoter regions and around the transcriptional start sites, results in the induction of gene expression. Therefore, primers were designed for both genes in the regions covering the promoter proximate to ATG and the region around the ATG (see Figure 4A). After crosslinking histones and DNA, and the fractionation of DNA and immunoprecipitation of those fragments loaded with H3K4me3 or H3K9Ac via specific antibodies, numerous fragments were analyzed using qPCR. Figure 4B shows the drought-inflicted changes in loading with these euchromatic marks at the different loci of *HvS40* and *HvNCED1* in WT and the overexpression line. Interestingly, in all samples from oeW1-2 plants, a decrease in loading with both euchromatic marks was observed around the ATG and in the proximate promoter region during drought treatment. This correlated with the decrease in the expression of *HvS40* and *HvNCED1* in both overexpression lines (Figure 2A) and indicates that WHIRLY1 is involved in epigenetic mechanisms controlling the expression of genes related to ABA biosynthesis.

## 3. Discussion

WHIRLIES are a family of small, plant-specific proteins, which intriguingly are localized to endosymbiotic organelles and the nucleus [16,17]. They are discussed to operate as a retrograde signal, and the transfer of WHIRLY1 from the chloroplast to the nucleus has been shown [25]. Many reports indicate a function of WHIRLY proteins in plants’ responses to biotic and abiotic stress [14,18,19,20,21,22,23], including nutrient deficiencies [38]. The mode of action of WHIRLY proteins is, however, still unclear. Interestingly, WHIRLY proteins are able to bind to ssDNA, and an interaction with the promoters of the transcription factor gene *WRKY53* and the senescence-associated gene *S40* has been shown [29,39]. ssDNA-binding proteins are known to act as positive or negative transcriptional regulators [40]. In fact, transcriptional regulation occurring either via interaction with transcription factors or via multifunctional chromatin modeling complexes involves the partial unwinding of double-stranded DNA and the formation of single-stranded loops [41,42,43,44]. In chloroplasts, WHIRLY proteins are a component of the TAC-complex associated with the nucleoids [26,27,28]. Nucleoids are multifunctional platforms integrating functions such as DNA replication, DNA repair and transcription, in connection to central biochemical processes, e.g., photosynthesis and signaling [45,46,47].

Our results indicate that the WHIRLY1 protein of *H. vulgare* acts as a regulatory factor of drought stress responses in the nucleus. The gain-of-function of WHIRLY1 delays the onset of drought-stress-induced premature senescence. In addition, the stress response is clearly affected, shown by the suppression of ABA biosynthesis genes *NCED*, *CCD5* and *CCD8* and of the ABA-dependent senescence and stress-related gene *HvS40*. This suppression of the drought-induced expression of genes involved in ABA biosynthesis consequently resulted in the suppression of ABA accumulation under drought conditions. Functional analysis of all differentially expressed genes revealed that further classes of genes were suppressed due to the gain-of-function of WHIRLY1. The data indicate that during drought stress, WHIRLY1 down-regulates transport processes, senescence processes, specific hormonal pathways and also some specific abiotic and biotic stress responses. On the other hand, under these conditions, bidirectionally functioning WHIRLY1 specifically up-regulates the genes involved in chloroplast development and function. This bidirectional function is typical for regulatory proteins balancing such trade-offs between abiotic and biotic stress responses and, in addition, developmental processes [48,49]. This sensitive and flexible balancing of different central pathways is obviously specific to particular stress and developmental situations, allowing for efficient acclimation to an ever-changing environment. Another environment will need other specific responses. Recently, it was shown that during growth under high light, the overexpression of WHIRLY1 shifts the trade-off between growth and resistance to the latter [30].

Possible binding sites of WHIRLY proteins in promotors have previously been identified. A 30-bp sequence in the promoter containing an inverted repeat sequence required for binding WHIRLY1 to the *PR10-a* promoter, that is, TGACANNNNTGTCA, has been named the elicitor response element (ERE) [19]. Later on, mutational analyses have shown that the GTCAAAAA/T sequence also conveys ERE activity [18]. This so-called PB (PBF-2 binding) element has been detected in the promoters of several defense genes and can overlap with the W-box (T/G)TGAC(C/T) and the TGACG element that are targets of the WRKY and TGA family transcription factors, respectively [50]. In barley, WHIRLY1 was found to bind to the promoter of the senescence-associated *HvS40* gene [29]. In accordance with the results on the downregulation of *HvS40* via overexpression (Figure 2A), the former study hypothesized that *HvS40* is negatively regulated by WHIRLY1 [29]. A bioinformatics search for further *cis*-acting elements in the barley genome known to be possible binding sites of WHIRLY proteins [17] showed that many genes found in this report to be up- or down-regulated due to the overexpression of *WHIRLY1* have these cis-elements in their promoter, supporting the idea that WHIRLY1 regulates the expression of these genes via physical interaction with cis-elements. These genes are listed in Appendix A. GO-enrichment analysis of up-regulated genes with the ERE *cis*-element revealed that many of these genes are involved in chloroplast function and primary metabolism (see Appendix A).

To further pinpoint the mechanism by which WHIRLY1 suppresses ABA-related stress signaling, we analyzed the epigenetic control of the two genes *HvNCED1* and *HvS40* by determining the loading of the promoter region and transcriptional start sites with the two euchromatic histone modification marks H3K4me3 and H3K9Ac. Our data showed that the overexpression of *WHIRLY1* alters these differential histone modifications during drought stress in a way where both euchromatic marks are withdrawn, which might result in a less open state and thereby in an observed decrease in transcription. Interestingly, a former publication showed that the knock-down of *WHIRLY1* via the RNAi approach also resulted in a delay in the drought-related onset of senescence and affected loading with euchromatic marks [14]. Here, we show that the overexpression of *WHIRLY1* causes similar effects as the knock-down, that is, a known pattern for scaffold proteins [51,52,53,54]. An optimum level of such scaffold proteins is needed for their platform function within multi-protein complexes. Both enrichment and loss of the protein might, under certain circumstances, result in a similar disturbance of the scaffold function.

Multi-protein complexes associated with single-stranded DNA loops in promoter regions are known to be a platform for the regulation of downstream target genes either via transcription factors or via chromatin remodeling processes [41,42,43,44]. Our results indicate that WHIRLY1 may act as a scaffold protein in these complexes. WHIRLY1 physically interacts with histone deacetylase HDA15, as shown by yeast two-hybrid, bimolecular fluorescence complementation and co-immunoprecipitation, and both proteins co-localize to the region near the transcription start site of some nutrient-recycling-related genes [55]. This led to the assumption that WHIRLY1 indeed acts as a repressor via recruiting HDA15, which causes the deacetylation of associated histones. The results presented here show that WHIRLY1 also causes the deacetylation of histones associated with ABA-related target genes. As illustrated in the working hypothesis scheme in Figure 5, WHIRLY1 could act as a retrograde signal when chloroplasts sense drought stress. Drought stress, as with other developmental and stress-related events, might lead to a transfer of WHIRLY1, which is present in the chloroplasts, to the nucleus, where it functions in regulatory complexes at the promoter and transcriptional start sites of various genes, among them being ABA-related genes such as *NCED*. Due to its location at regions of un-winded, single-stranded DNA, WHIRLY1 might act as a scaffold for the docking of multiple proteins, e.g., transcription factors and epigenetic regulators, such as histone deacetylases. This results in the cleavage of acetyl residues and eventually in a decrease in the transcription of these genes. Through this series of events, WHIRLY1 may act as a plastidial signal-damping ABA-related stress response. Similar negative feedback loops have already been described in ABA signaling [56,57]. Such function as a platform which in response to different environmental conditions dynamically interacts with different regulatory factors could allow balancing major developmental and stress-related pathways in an environment-sensitive manner. 

## 4. Material and Methods

### 4.1. Plant Materials, Growth Conditions and Drought Treatment

Seeds of barley (*Hordeum vulgare* L. cv. Golden Promise) wild type (WT) and overexpression lines (oeW1-2 and oeW1-15) [30], in which the expression of *WHIRLY1* is driven by the constitutive Ubiquitin promoter from maize (*ZmUBI1-int*), were stratified on wet paper for 72 h at 4 °C and kept for an additional 24 h at room temperature in the dark. After germination, barley seedlings were transferred into soil (9 plants/pot). Each pot containing 2.7 kg ED73 soil (Einheitserdewerk GmbH, Sinntal, Germany) was placed into a phytochamber under standardized conditions (16 h light intensity of 300 µE·m^−2^·s^−1^ and 8 h dark at 25 °C). To induce drought stress, irrigation was stopped at 11 days after sowing (das), following the method previously described in [58]. In the control pots, water capacity was kept at 65% throughout the whole experiment.

### 4.2. Physiological Measurements

Measurements of physiological parameters (relative chlorophyll content and photosystem II (PS II) efficiency) on leaf material were performed using primary foliage leaves collected in the middle of the light period. Maximum values were set as 100%. Data were obtained from at least three independent experiments comparing WT and overexpression plants under control and drought conditions.

#### 4.2.1. Relative Chlorophyll Content

The relative chlorophyll content per unit leaf area was measured using a SPAD (soil plant analysis development) analyzer (Konica Minolta Sensing Europe B.V., Munich, Germany). The transmittance of red (650 nm) and infrared (940 nm) radiation was measured noninvasively at intact primary leaves. Relative SPAD values are related to the chlorophyll content in a linear manner [59]. Each data point represents the average of at least three independent biological experiments with measurements at the middle of primary leaves of 9 independent plants in each experiment.

#### 4.2.2. PS II Efficiency

The PS II efficiency (F_V_/F_M_), as a measure of the quantum yield of PS II, was measured via chlorophyll fluorometry using a photosynthesis yield analyzer (Mini-PAM, Walz GmbH, Effeltrich, Germany) via the pulse amplitude modulation (PAM) method [60]. Measurements were executed at the mid position of the primary leaves after 15 min dark adaptation. Each data point represents the average of at least three independent biological experiments with measurements in the middle of primary leaves of 9 independent plants in each experiment.

### 4.3. Gene Expression Analysis

Total RNA was isolated with Trizol from the primary leaves of barley WT and overexpression lines grown under control and drought treatments as described earlier [14,61]. To prevent contamination by genomic DNA, RNA samples were treated with RNase-free DNaseI (Roche-Diagnostics GmbH, Mannheim, Germany). Synthesis of the first-strand cDNA was performed by reverse transcription of 1 µg of total RNA with RevertAid H Minus Reverse Transcriptase (Thermo Fisher Scientific Inc., Waltham, MA, USA) in a volume of 20 µL as described in the operating manual. The qRT-PCR reactions were performed in 10 µL containing 5 µL KAPA SYBR^®^ Fast mastermix (KAPA Biosystem, Cape Town, South Africa), 0.2 µM of each gene-specific primer (Appendix A) and 2 µL of diluted (1:64) template cDNA. PCR was executed in a MyiQ™2 system (Bio-Rad Laboratories Inc., Göttingen, Germany). The relative expression of genes of interest (GOI) was calculated using a modified DeltaDeltaCT method. Thereby, the expressions of GOI of the control (ΔCt_Control_ = Ct_GOI_ − Ct_Ref_) and drought-stressed (ΔCt_Drought_ = Ct_GOI_ − Ct_Ref_) plants were normalized to the expression of reference genes (*HvEF1a*: HORVU.MOREX.r2.5HG0427750 and *HvPP2A*: HORVU.MOREX.r2.4HG0335530), which showed a stable Ct value in both drought and control conditions. The values were further normalized to the expression of the gene of interest in control plants at the same time point ΔΔCt = ΔCt_Drought_ − ΔCt_Control_. Each data point represents the average of measurements of three independent biological replicates.

### 4.4. Transcriptome Analysis

Total RNA was extracted from the primary leaves of WT and the oeW1-2 line grown for 28 days under control or drought conditions, as described above. The integrity of RNA was checked via the Agilent 2100 Bioanalyzer system with RNA 6000 pico kit (Agilent, Santa Clara, CA, USA). Samples of a high quality (A_260/280_ = 1.8–2.2, ≥50 ng/μL, RIN ≥ 6) were sent to Novogene (Cambridge, UK) for Illumina sequencing, including mRNA enrichment and library preparation. The clean raw reads (no adapter sequences, no unidentified bases, high Phred score) were mapped into the genome of barley cv. Morex (version 2) as reference [62] using HISAT2 [63], and assembled using StringTie [64]. Afterward, the mapped read counts that correlated with the gene expression level were quantified as FPKM (fragments per kilobase of transcript sequence per million base pairs sequenced) using the Subread package [65].

The differential expression (DE) analysis between each pair of samples with 3 biological replicates was performed using the R/Bioconductor package *DESeq2* [66,67]. Genes with log_2_(FoldChange) ≥ 1 and adjusted *p* ≤ 0.05 were defined as being significantly different. The clustering analysis and the heatmap of the gene expression level were generated using the Pheatmap package (version 1.0.12).

The gene functional enrichment analysis was performed using the TRAPID platform [31]. The significance of over-representation was determined using the hypergeometric distribution, and the Benjamini and Hochberg method was applied to correct for multiple testing.

### 4.5. Chromatin Immunoprecipitation (ChIP) Analysis

ChIP was performed as described by [68,69] with modifications. One g of primary leaves was harvested at 27 days and cross-linked with formaldehyde (0.4 M sucrose, 10 mM Tris-HCl (pH 8), 1 mM Na-EDTA, 1 mM PMSF and 1% formaldehyde [70]) for 10 min at 4 °C via vacuum infiltration (18–20 mmHg). The cross-linking reaction was then stopped by adding glycine to a final concentration of 0.1 M followed by another 5 min of vacuum infiltration. To extract chromatin, the nuclei were isolated and re-suspended in nuclei lysis buffer (50 mM Tris-HCl (pH 8), 10 mM Na-EDTA, 1% SDS, 0.1 mM PMSF and Protease Inhibitor Cocktail; Roche, Mannheim, Germany). Chromatin was sheared via sonication using a Covaris M220 Focused Ultrasonicator (Woburn, MA, USA) to an average fragment size of 300 bp, which was checked via gel electrophoresis (1% TAE agarose gel) for each biological replicate.

The DNA immunoprecipitation was performed via incubation with Dynabeads™ Protein G (Invitrogen, Waltham, MA, USA) coated with antibodies against H3K4me3 (ab8580) and H3K9ac (ab10812) obtained from Abcam (Cambridge, UK) at 4 °C overnight. In addition, input DNA control (representing the total DNA) and no-antibody control (mock) were prepared. To isolate and clean the immunoprecipitated DNA, the amount of precipitated DNA was quantified via qPCR. The qPCR reactions were performed in a total volume of 10 µL containing 5 µL KAPA SYBR^®^ Fast master mix (KAPA Biosystem, Cape Town, South Africa), 0.2 µM of each gene-specific primer (see Appendix A), 10 µM fluorescein (Bio-Rad Laboratories Inc., Göttingen, Germany) for well-factor calibration and 5 µL precipitated template DNA. To exclude the amplification of unspecific products, no-template controls were carried out. PCR was executed using the MyiQ™2 System (Bio-Rad). Mock values, which were below 0.1% of input DNA, were subtracted from IP samples. Subsequently, drought-stressed (D) results were divided by the results of untreated mature control (C) primary leaves (ratio D/C) for WT and oeW1 lines, respectively.

### 4.6. Quantification of Abscisic Acid (ABA)

The content of ABA was quantified using a standardized ultra-performance liquid chromatography–tandem mass spectrometry (UPLC-MS/MS)-based method according to Balcke et al. [71]. About 50 mg of leaf material was extracted with 500 µL methanol supplied with [^2^H_6_] ABA (OlChemIm, Olomouc, Czech Republic) as the internal standard. After centrifugation, the supernatant was diluted with nine volumes of water and subjected to solid phase extraction on HR-XC (Chromabond, Macherey-Nagel, Düren, Germany). Elution was conducted with 900 µL acetonitrile, and 10 µL of the eluate was subjected to UPLC–MS/MS. The ABA content was calculated using the ratio of analyte and internal standard peak heights.

## Figures and Tables

**Figure 1 ijms-24-06326-f001:**
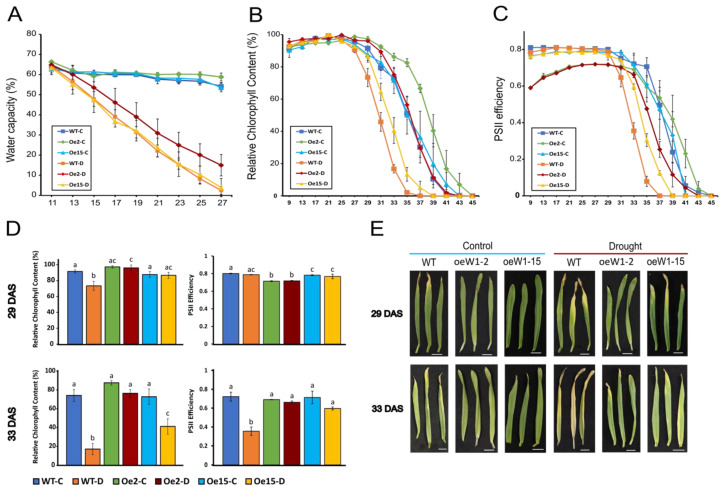
Changes in chlorophyll content and photosystem II efficiency in wild-type barley (*Hordeum vulgare* cv. Golden Promise) (WT) and two *WHIRLY1* overexpression lines (oeW1-2 and oeW1-15) under control (C) and drought (D) conditions (irrigation stopped at 11 das). (**A**) Soil water capacity. (**B**,**C**) Relative chlorophyll content (SPAD values) and PS II efficiency (F_V_/F_M_) during drought treatment. (**D**) Relative chlorophyll content (SPAD values) and photosystem II efficiency (F_V_/F_M_) at 29 and 33 das. Different letters above the graph bar designate statistically significant differences according to two-way ANOVA using Tukey’s HSD test with *p* < 0.05. (**E**) Senescence phenotype of the primary leaves from WT and oeW1 barley lines. Bars represent 1 cm. (**A**–**D**) Error bars show the standard deviation (SD) of the corresponding value.

**Figure 2 ijms-24-06326-f002:**
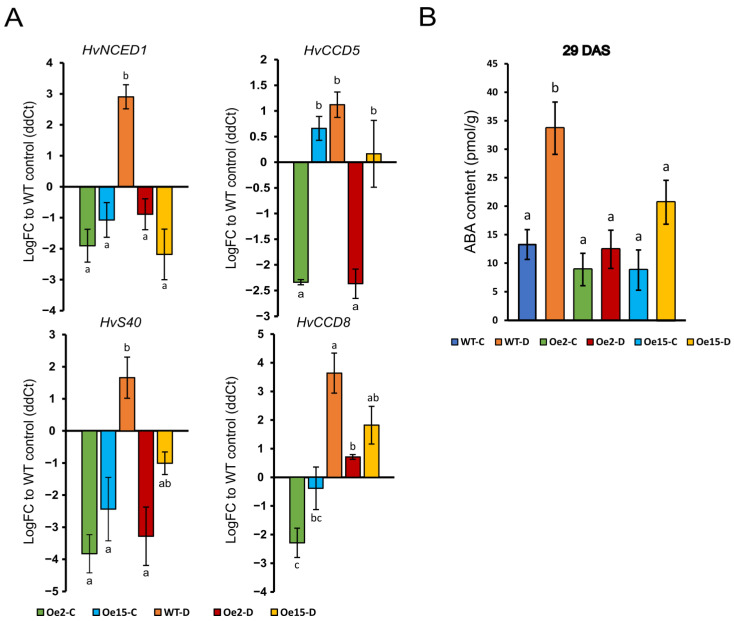
Expression of ABA biosynthesis-related genes and ABA levels in drought-stressed primary leaves (28 das) of WT and *WHIRLY1* overexpression lines 2 and 15 compared to leaves of control plants. (**A**) Changes in relative expression level of senescence- and stress-related gene *HvS40* and of ABA biosynthesis-related genes (*HvNCED1*, *HvCCD5* and *HvCCD8*) measured via qPCR. Each bar represents the mean value from 3 replications, and the error bar shows the SEM. (**B**) ABA concentration of primary leaves at 29 das. Each bar represents the mean value from 3 separate replicates, and the error bar shows the standard error of the mean (SEM). Different letters above the graph bars in A and B designate statistically significant differences according to two-way ANOVA using Tukey’s HSD test with *p* < 0.05.

**Figure 3 ijms-24-06326-f003:**
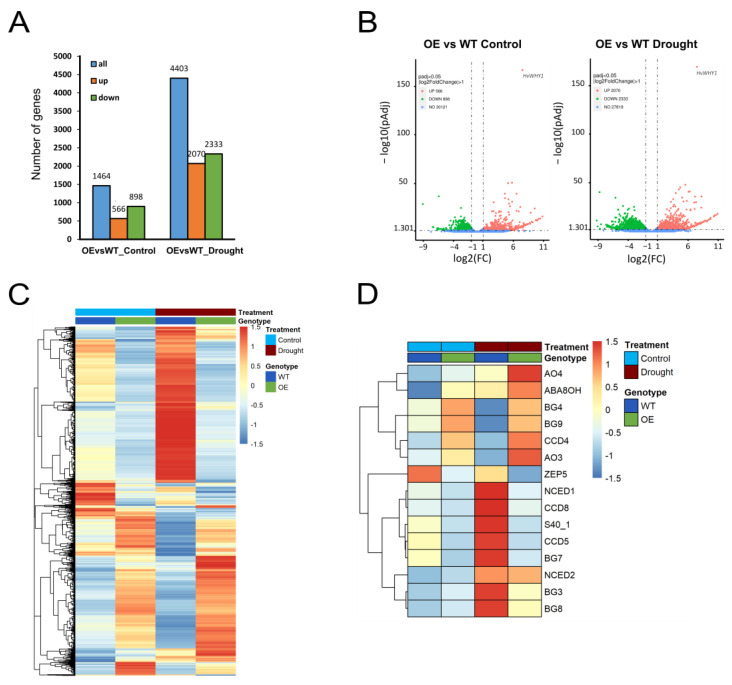
RNA sequencing of primary leaves at 28 das from WT and oeW1-2 grown in control and drought conditions. (**A**) The numbers of differentially expressed genes (DEGs). (**B**) Volcano plots showing distribution of differentially up- and down-regulated genes under control and drought conditions. (**C**) Z-score heatmap of DEGs generated from FPKM value. (**D**) Expression of ABA biosynthesis-related genes. The heatmap was generated from the RNA sequencing data with manual curation. The bars in (**C**,**D**) show color codes for the heatmap Z-score scale.

**Figure 4 ijms-24-06326-f004:**
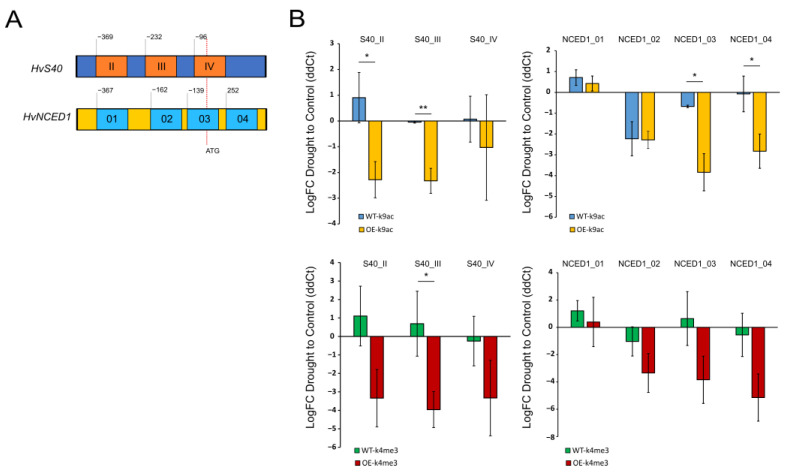
ChIP-qPCR analysis of drought-stress-related loading of ATG and the proximal promoter region of *HvS40* and *HvNCED1* with euchromatic marks H3K4me3 and H3K9Ac in the leaves of oeW1-2. (**A**) Schematic illustration of ATG and promoter region of *HvS40* and *HvNCED1*, indicating sectors determined by qPCR. (**B**) ChIP qPCR of the investigated regions showing the drought-stress-induced change in loading with euchromatic marks H3K4me3 and H3K9Ac compared to controls. Each graph bar represents the average value from 3 separate replications, and the error bar shows the SEM. The asterisks above the graph bar show statistically significant differences according to Student’s *t*-test with * *p* < 0.05 and ** *p* < 0.01.

**Figure 5 ijms-24-06326-f005:**
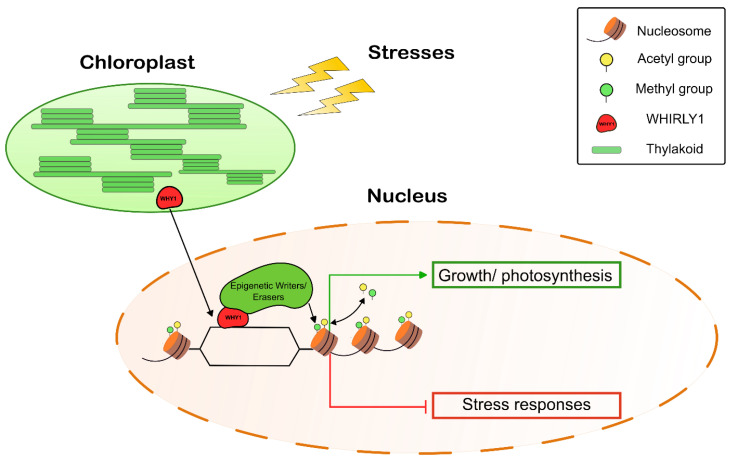
Hypothetical working model of WHIRLY1, moving from the chloroplast to the nucleus during drought stress and acting as a platform regulating gene expression via epigenetic mechanisms.

## Data Availability

The data presented in this study are available in article and Appendix A.

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
