# Peer review of "WHIRLY1 Acts Upstream of ABA-Related Reprogramming of Drought-Induced Gene Expression in Barley and Affects Stress-Related Histone Modifications"

_ijms, 2023, doi:10.3390/ijms24076326_

Round 1

Reviewer 1 Report

In my opinion, this is an interesting job, done at an excellent level. The obtained data and conclusions give the impression of being reliable.

From the comments—just not quite an accurate list of references.

Author Response

Thanks for the great review!

  • We now cross-checked all references and accurately list them

Reviewer 2 Report

In this manuscript, Minh Bui Manh (ijms-2288346) and colleagues studies the WHIRLY1 gene function in barley.

This article was really interesting for the reviewer.  The article was well written, figures and data represent one unit.  I have no question and suggestion for the authors. I can accept this paper in present form for publication.

1. What is the main question addressed by the research?
The main question was, how changes the gene expression pattern, during the drought stress in the WHIRLY1 over expression plants in barley. The authors choose the first barley leaves to do different studies, gene expression, chlorophyll content and PSII activity. And after the RNA seq analysis, the authors verify the promoter binding by CHIP.

2. Do you consider the topic original or relevant in the field? Does it
address a specific gap in the field?
The topic was in this special issue: "Epigenomics and Crop Improvement"  this research is cover the topic, because the article based on to improve barley stress tolerance.

3. What does it add to the subject area compared with other published
material?
This article is studied the WHIRLY1 gene function in drought stress in barley. In other published article, did not studied this possible function.

4. What specific improvements should the authors consider regarding the
methodology? What further controls should be considered?
In my opinion, that I addressed this question. I checked carefully the article.

5. Are the conclusions consistent with the evidence and arguments presented
and do they address the main question posed?
My answer is yes, that’s why I recommend this article for publication, without any change.

6. Are the references appropriate?
In my opinion, the references are correct. The article has 73 references: 48 references from the last 10 years, and from the older ones, most of it basic methodological or review article.

7. Please include any additional comments on the tables and figures.
In the article I found the figures in the right place, the figure legends were written correctly. I have no problem with the figure quality.  The article has a Supplemental Figures and Tables legends too.

Author Response

Thanks for the great review!

No specific points changed.

Reviewer 3 Report

The manuscript entitled "WHIRLY1 acts upstream of ABA-related reprogramming of drought-induced gene expression in barley and affects stress-related histone modifications" compared drought response, gene expression and transcriptome analysis between the wild type and WHIRLY1 overexpression line to understand the function and possible mechanism of WHIRLY1 in the drought response.  The research was conducted properly with clear conclusions were drawn from obtained data. I would recommend accepting the manuscript for publication, but some minor revisions will be needed:

1. in Abstract, the sentence lines 13-14 could be deleted. sentences in lines 27-28 make nonsense.

2. The introduction is the part needs more revisions. It includes a quite more proportion for role of ABA and ABA-biosynthesis, but much less background about WHIRLY1. However, the main focus of this research is WHIRLY1 rather than ABA. I would recommend moving contents in the discussion part about WHIRLY1 to the introduction part, to talk about the status and function of WHIRLY1, and then to mention why the research was conducted and the objectives of the study. 

3. results part mostly are fine, Fig 3C and D should notify more about the numbers 1.5, 0.5, 0, -0.5, .. are they Log2(folder changes)? 

4. Discussion: according to the proposed hypothesis about working model of WHIRLY1, does that mean the WHIRLY1 proteins were synthesized and then transported into and stored in chloroplast? If that's true the plant stores WHIRLY1 proteins even though they were not under stresses.

Author Response

Thanks a lot for the helpful review!

  • We deleted the first two sentences (lines 13-15) in Abstract.
  • Lines 28-30 in Abstract were changed to prevent misunderstanding.
  • Following the helpful suggestions of the reviewer, we shortened the part describing the role of ABA, and moved some more information about WHIRLY proteins from Discussion to Introduction. (Changes in lines 57-70, 79-93, 249-253, 262-264).
  • We added information about color code in legend to Figure 3.
  • The reviewer addresses an interesting point: WHIRLY 1 is present in chloroplasts also under control conditions. It is acting there in the TAC complex (as described in the text in Introduction and Discussion). Our model based on our results proposes that during drought it is transferred to the nucleus where it regulates stress-related gene expression. In order to better implement this point in the Discussion, we changed the text (lines 332-334) and included that WHIRLY protein is in chloroplast and that drought, as also other developmental and stress conditions, might lead to a transfer from chloroplast to nucleus.